# The impact of whole genome duplications on the human gene regulatory networks

**Francesco Mottes** [1], **Chiara Villa** [2], **Matteo Osella** [1], **Michele Caselle** [1] *

**1** Department of Physics, University of Turin & INFN, Turin, Italy, **2** School of Mathematics and Statistics, University of St Andrews, Mathematical Institute, North Haugh, St Andrews, United Kingdom

* caselle@to.infn.it

**Data Availability Statement:** The raw data used for this study are all publicly available from their respective sources. The data and the code required to replicate the analyses and figures in this work are available on Zenodo with the following DOI: 10.

## Abstract

This work studies the effects of the two rounds of Whole Genome Duplication (WGD) at the origin of the vertebrate lineage on the architecture of the human gene regulatory networks. We integrate information on transcriptional regulation, miRNA regulation, and protein-protein interactions to comparatively analyse the role of WGD and Small Scale Duplications (SSD) in the structural properties of the resulting multilayer network. We show that complex network motifs, such as combinations of feed-forward loops and bifan arrays, deriving from WGD events are specifically enriched in the network. Pairs of WGD-derived proteins display a strong tendency to interact both with each other and with common partners and WGD-derived transcription factors play a prominent role in the retention of a strong regulatory redundancy. Combinatorial regulation and synergy between different regulatory layers are in general enhanced by duplication events, but the two types of duplications contribute in different ways. Overall, our findings suggest that the two WGD events played a substantial role in increasing the multi-layer complexity of the vertebrate regulatory network by enhancing its combinatorial organization, with potential consequences on its overall robustness and ability to perform high-level functions like signal integration and noise control. Lastly, we discuss in detail the RAR/RXR pathway as an illustrative example of the evolutionary impact of WGD duplications in human.

## Author summary

Gene duplication is one of the main mechanisms driving genome evolution. The duplication of a genomic segment can be the result of a local event, involving only a small portion of the genome, or of a dramatic duplication of the whole genome, which is however only rarely retained. All vertebrates descend from two rounds of Whole-Genome Duplication (WGD) that occurred approximately 500 Mya. We show that these events influenced in unique ways the evolution of different human gene regulatory networks, with sizeable effects on their current structure. We find that WGDs statistically increased the presence of specific classes of simple genetic circuits, considered to be fundamental building blocks of more sophisticated circuitry and commonly associated to complex functions. Our

5281/zenodo.5110112 Our processed lists of SSD and WGD paralogues and the processed regulatory networks are also easily downloadable from the following GitHub repository: https://github.com/fmottes/wgd-network-motifs.

**Funding:** This work was in part supported by the "Departments of Excellence 2018–2022" Grant awarded by the Italian Ministry of Education, University and Research (MIUR) (L.232/2016). In particular, authors FM, MO and MC benefitted from it. The funders had no role in study design, data collection and analysis, decision to publish, or preparation of the manuscript.

**Competing interests:** The authors have declared that no competing interests exist.

findings support the hypothesis that these rare, large-scale events have played a substantial role in the emergence of complex traits in vertebrates.

## Introduction

Gene duplication is one of the main drivers of evolutionary genomic innovation [1–3]. Small Scale Duplications (SSDs) typically involve a single gene or a small set of genes within a well defined genomic locus. More rarely, a large-scale genomic duplication may occur, which involves a macroscopic portion of the genome. Such events are called Whole Genome Duplications (WGDs), and it is by now clear that they played a major role in evolution [4, 5]. SSD events can induce a local exploration of the phenotypic landscape, introducing small and incremental changes to the genome and consequently to the cell functions. WGD events, on the other hand, typically entail more sudden and dramatic phenotypic changes. They also most likely produce immediate dire consequences on the fertility and fitness of the organism that comprimise its short-term survival [6]. As a result, most WGD events are not fixated in the population. In some peculiar circumstances, though, they can constitute an immediate evolutionary advantage and help reducing the risk of extinction of the affected lineage [7]. Moreover, increasing evidence points towards a central role of WGD in the successful response to the stress induced by sudden environmental changes [5]. Also, fixated WGD events can boost the biological complexity of the organism in the long term [5].

This paper focuses specifically on the human genome, and thus on the two rounds of WGDs that occurred about 500–550 Millions of years ago. More than 50 years ago Susumu Ohno [1] proposed in a seminal paper that two rounds of WGD were at the origin of the vertebrate lineage. The hypothesis was met with both interest and skepticism, and it was only the advent of high-throughput sequencing that provided reliable evidence supporting ancient WGD events. In 1997, a WGD event was unambiguously detected for the first time in Saccharomyces cerevisiae [8, 9] and a few years later in Arabidopsis thaliana [10]. Finally, in 2005 Ohno's original intuition regarding the two WGD events at the origin of the vertebrate lineage was also confirmed [11], and WGD duplicates are now also called "ohnologs" in his honour. These events are conjectured to have played a central role in the evolution of complex traits associated with vertebrates. For example, a multi-omics analysis of the Amphioxus genome has shown that the two rounds of vertebrate WGD significantly increased the complexity of the vertebrate regulatory landscape, and possibly boosted the evolution of morphological specializations [12]. It was also shown that an important class of human highly interacting proteins, involved in processes that are crucial for the organization of multicellularity, was mainly created by vertebrate WGD [13]. On a more general ground, WGD events are recognized to have played a major role in the introduction of evolutionary novelties in many species, by influencing gene retention and selection, dosage balance and subgenome dominance effects among others. This phenomena are especially well studied in plants, in particular in A. thaliana [14–16].

The identification of WGD pairs or quartets in vertebrates is a highly non trivial task because of their ancient origin [17]. In fact, a stable and reliable list of human WGD gene pairs was only recently proposed [18–20]. This advance made it finally possible to analyze the evolutionary role of WGD and SSD also in human. As a consequence, few interesting features have been identified to be uniquely associated to WGD pairs. For example, WGD genes are subject to more stringent dosage balance constraints and are more frequently associated with disease with respect to other genes [21]. Moreover, WGD genes are threefold more likely than non

WGD ones to be involved in cancers and autosomal dominant diseases [18]. This observation led to the suggestion that WGD genes are intrinsically "dangerous", in the sense that they are more susceptible to dominant deleterious mutations than other genes [22]. From a functional point of view, WGD genes are more frequently involved in signalling, development and transcriptional regulation and they are enriched in Gene Ontology categories generically associated to organismal complexity [18, 19, 23–25]. From the gene expression point of view, both the gene expression profile and the subcellular localization seem to be more divergent between the two partners of a WGD-derived pair than for gene pairs derived from SSD [19]. In the same work, the authors also note that WGD-derived genes contain a larger proportion of essential genes than the SSD ones and that they are more evolutionary conserved than SSD. Remarkably, several of these recent observations on vertebrates WGD genes agree with what was found years ago both in yeast [26] and in A. thaliana [24, 27]. This "universality" supports the hypothesis of general principles or mechanisms behind the unexpected retention of WGD genes and their interactions.

The goal of the present work is to pinpoint the different roles played by the two types of gene duplications—SSD and WGD—in shaping the architecture of the human gene regulatory network. In particular, we investigate the local structure—mainly by analysing the network motif enrichments—of the transcriptional regulatory network, the protein-protein interaction network and the miRNA-gene interaction network, which are partially represented in Fig 1A, 1B and 1C respectively. Network motifs are statistically enriched subgraphs that can be found in many complex networks [28] and they assume particular significance in biology and for gene regulatory networks in particular. In fact, in this context network motifs identified gene circuits that can perform relatively simple computations with specific biological functions. These simple modules can then assemble into a larger network to implement complex and robust regulatory strategies [29]. As shown in Fig 1E, gene duplications—and WGD in particular—can create motifs in a very straightforward way by duplicating the genes involved in a simple regulatory interaction. Even though this is certainly not the only way in which motifs may be created, we expect duplication events to have a major impact on the creation and, most importantly, the subsequent retention of these local structures.

We therefore analyzed the statistical enrichment of a selection of motifs—represented in Fig 1D—whose functional importance is widely recognized [29]. We observe that SSD and WGD gene pairs are statistically over-represented in different types of motifs. This result is in general agreement with previous observations on the yeast transcriptional network [30]. We will show that also the structure of additional layers of regulation present in the human genome, such as miRNA regulation, has also been influenced by duplication events. In conclusion, this work shows that SSD and WGD events shaped the multiple layers of regulation in the human genome in different ways and jointly contributed to their current structure. The specific consequences of WGD events on the regulatory network seem to be associated to an increased redundancy and complexity that would be hard to attain through a sequence of small-scale events.

## Materials and methods

### Small-scale and whole-genome duplicates

**WGD paralogues.** The WGD gene pairs were obtained by merging the results of *Makino and McLysaght* [21] with the latest available OHNOLOGS database [20]. In order to have a high-confidence list of paralogies, we considered only WGD couples corresponding to the *strict* criterion in the OHNOLOGS database. Moreover, all the couples that were not recognized as paralogues in the current version of the Ensembl database were excluded. To ensure

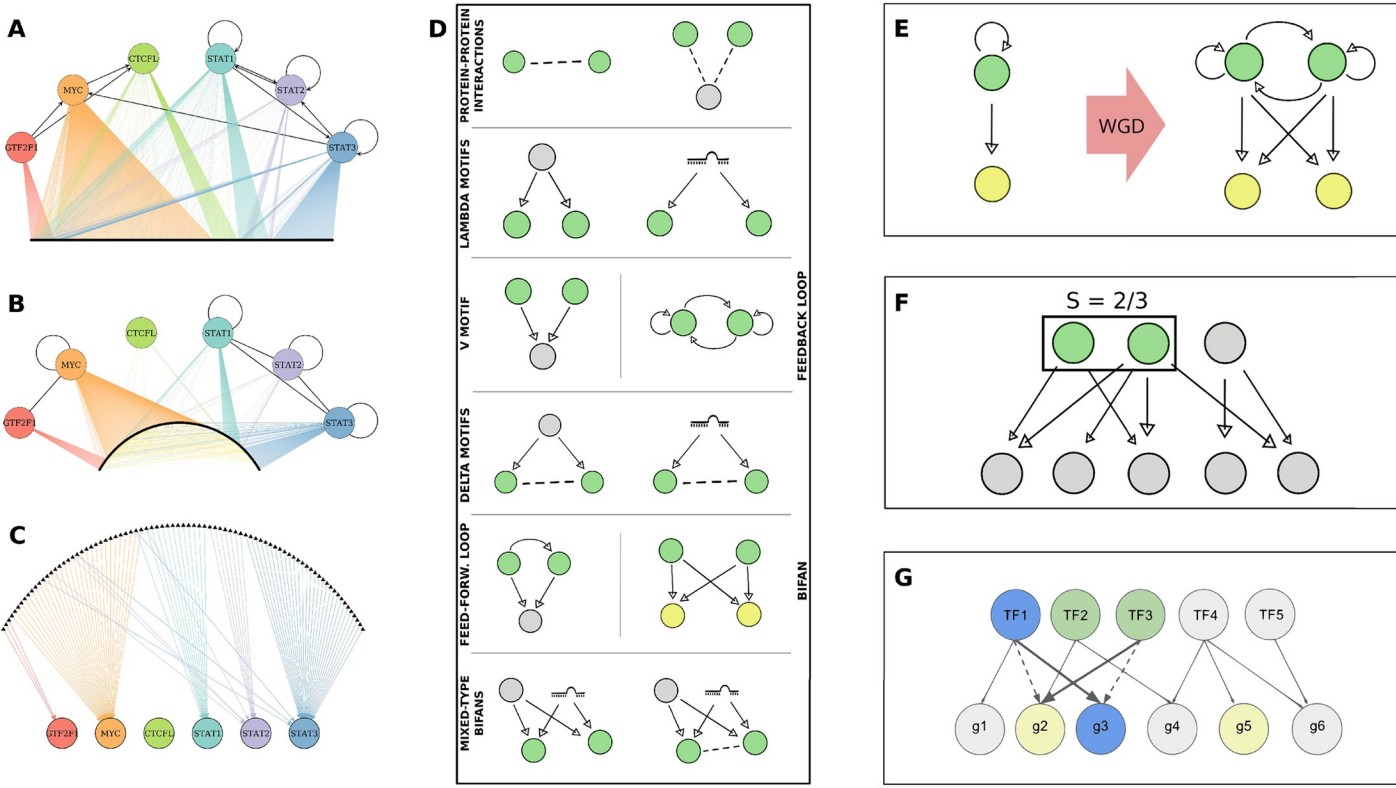

**Fig 1. Regulatory networks and network motifs.** Interactions involving an illustrative subset of TFs are shown on the left for each of the regulatory mechanisms studied in the present work, i.e. for **(A)** transcriptional regulations from the ENCODE network, **(B)** protein-protein interactions in the PrePPI network, and **(C)** miRNA-gene regulatory interactions in the TarBase network. TFs are represented as colored circles, target genes as small black dots (here appearing as a thick black lines due to large number of genes), and miRNAs as black triangles. Black lines indicate interactions between TFs, while other interactions have the color of the involved TF. Yellow lines are interactions between non-TFs. **(D)** An overview of the network motifs that will be considered in the following. Gray circles represent generic genes, same-color circles (green or yellow) are paralogues, and miRNAs are represented in a stylized form. Solid arrows represent regulatory interactions, while dashed lines represent protein-protein interactions. **(E)** Illustration of how a WGD event can easily create FFLs and Bifans by duplicating the components of a simple regulatory interaction in which the regulator also has self-regulation. Many of the created interactions will then be lost during the evolutionary process, leaving only those that are not negatively selected. **(F)** Example of the structure of a Dense Overlapping Regulon (DOR) embedded in a gene regulatory network, with the target similarity $S$ calculated for an illustrative couple. **(G)** Graphical representation of the degree-preserving procedure used to generate the null models: the dashed links are randomly chosen and their ends swapped, thus generating the new bold links. Note that all of the involved genes maintain their in and out degree in the process.

full compatibility among all of the datasets employed, we updated the gene names to the latest officially accepted version—data about the status of gene names were obtained from the HGNC online service [31]. Finally, only protein-coding genes (according to the Ensembl database) were considered in our list of paralogues. With these restrictions, we ended up with a list of 8070 WGD-derived paralogue couples, comprising 7324 different genes.

**SSD paralogues.** SSD-derived paralogues were obtained from the list of all human paralogues involving protein-coding genes in the Ensembl database [32], and subtracting from this list all of the couples that were previously identified as derived from a WGD. One additional factor that must be taken into account when dealing with the distinction between WGD and SSD couples is the huge spread of duplication ages of the SSD paralogues. The two rounds of WGD happened relatively close in time, approximately around the appearance of the Vertebrate lineage $\sim$ 500 Mya. Given this timescale, it is reasonable to assume that the currently retained WGD gene couples have experienced similar evolutionary forces (neutral or selective). On the other hand, SSD couples are continuously generated throughout the history of the human genome evolution. Therefore, there can be SSD events that are significantly more

recent than the two rounds of WGD. Following this recent events, sequence evolution had relatively less chances to modify and rewire the gene interactions involving the resulting paralogues. Therefore, in order to make a sensible comparison between SSD and WGD couples, it is necessary to rule out possible confounding effects due to the different ages of paralogues. Such effects are indeed present, as we show in detail in Fig B in S1 Text. The duplication age of paralog gene couples was estimated by considering their most recent common ancestor. Specifically, we considered SSD couples whose most recent common ancestor is older than *Sarcopterygii* as roughly contemporary to WGD couples. This approach is in line with a previous suggestion [18] and indeed the estimated ages are compatible, as shown in S1 Text (Fig A). With these criteria, we identified 8663 young SSD duplicates (comprising 3442 genes) which we excluded from the comparison, and a final list of 13,618 SSD genes organized in 122,889 gene couples that we can safely use for a comparison with WGD genes couples.

**Non-duplicated gene couples.** In the analyses that follow we will sometimes compare the results obtained for SSD and WGD couples to those obtained for non-duplicated gene couples. We consider as non-duplicated couples all of the couples that one can construct with the genes in a specific network that are neither SSD couples nor WGD couples.

## Transcriptional regulatory network

We used the human transcriptional regulatory network presented in [33], a portion of which is displayed in Fig 1A. The network was obtained by the curation of data from ChIP-seq experiments by the ENCODE project, so we will be referring to it in the following as the "ENCODE network". We combined the information regarding proximal and distal regulation into a single regulatory network, with 122 transcription factors (TFs) and 9986 target genes. ChIP-seq based transcriptional networks should present the least amount of biases for the kind of analysis we are interested in, which essentially focuses on duplicated genes and network motifs. In fact, there are essentially three other methods to construct transcriptional regulatory networks besides Chip-seq derived networks (see for instance [34] for a recent review). Literature-based collections (such as TRRUST [35] or HTRIdb [36]) are by definition biased towards genes that received more attention from the scientific community. As pointed out in the Introduction, WGD-derived genes were shown to be often associated with diseases and organismal complexity, which are preferential subjects of published papers. Another possible approach is based on *in silico* predictions of the interactions from TF binding sequences. However, many of the duplicated TFs (especially the recent ones) can still present very similar binding sequences. Therefore, a network constructed in this way would lead to an artificially strong enrichment of some motifs (e.g., *V* motifs, shown in Fig 1D). Finally, methods based on reverse engineering gene expression data, such as the popular ARACNE [37], involve a pruning step that leads to an artificial decrease of the network clustering coefficient, and thus to an alteration in the statistics of three-node motifs.

## Protein-protein interaction networks

We extracted the protein-protein interaction (PPI) network from the PrePPI database [38] and the STRING database [39]. We downloaded the high-confidence predictions from the PrePPI database, selecting only the experimentally validated interactions, and updated the gene identifiers. The result is a network of 15,762 genes and 237,272 PPIs. From the STRING database, we selected interactions that were both experimentally validated and with high confidence score (interaction score > .700, a parameter pre-set by the authors), in order to enforce stringency and to have a network size comparable with the size of the PrePPI network. We ended up with a STRING PPI network with 10,725 genes and 108,129 PPIs. There is a large

overlap in the nodes present in the two networks (10,087 genes are in common) but a much lower overlap in the interactions (only 36,863 interactions are present in both networks). We will present in the main text the results obtained with the PrePPI network (a portion of which is shown in Fig 1B). However, all of the results are confirmed by analysis of the STRING network (see S1 Text, Figs D and E), thus proving the robustness of our results.

### miRNA-gene interaction networks

The miRNA-target interaction networks we considered come from the TarBase database [40] and the mirDIP database [41]. The TarBase network was constructed by selecting all the interactions coming from normal (non-cancer) cell lines or tissues, with positive evidence for a direct interaction between the miRNA and the target gene. This leaves us with 913 miRNAs regulating 10,497 genes, with 89,736 interactions. The mirDIP database integrates instead miRNA-target predictions coming from different databases and prediction methods, combining the different database-specific scores into a unified integrative score. Since no specific method is provided in order to choose an integrative score threshold, we chose to keep the 90,000 top-scoring interactions. Such a stringent threshold allows us to make a sensible comparisons with the TarBase network. The resulting mirDIP network has 513 miRNAs and 7965 genes with 89,991 interactions. As for the PPI networks, the overlap between the nodes is very high (406 miRNAs and 6241 genes are in common), but the overlap in edges is pretty low (only 9320 interactions are found in both networks). In the rest of the paper, results obtained with the TarBase network will be shown (represented in Fig 1C). The analogous results obtained with mirDIP network are available in S1 Text (Fig F). Again, the trends we find are robust despite the low overlap between the two networks.

### Network motifs

Networks motifs are combinations of nodes and regulatory interactions which are statistically over-represented in the regulatory network, with respect to an ensemble of null network models. They were shown to perform elementary regulatory functions [29] and the common lore is that some motifs were positively selected for by evolution precisely because of their ability to perform elementary computations. Such elementary modules can then be composed together to implement more complex regulatory functions in the regulatory network [42]. This paper focuses on network motifs involving pairs of duplicated genes, as illustrated in Fig 1D.

Two duplicated transcription factors may regulate the same target (or set of targets) without interactions between the two duplicated genes, in a configuration we refer to as V motif. On the contrary, a couple of genes may be regulated by the same TF or by a common miRNA, giving rise to a Λ motif. We will explicitly distinguish between transcriptional and miRNA-mediated Λ motifs. If the duplicated genes involved in a Λ motif also interact at the protein level, we have a Δ motif, which again can be transcriptional or miRNA-mediated. The duplicated genes may be simultaneously involved in transcriptional and miRNA-mediated Λ or Δ motifs, hence resulting in mixed-type network motifs. More complex transcriptional motifs will also be analyzed, such as feed-forward loops (FFL) and feedback loops (FBL), including self-regulations and toggle-switch-like architectures. We will also consider Bifan motifs, where a couple of duplicates regulates another one but there are no interactions between the two regulators, and FFL+Bifan motifs, which have the additional regulatory interaction between one regulator and the other. Finally, we will also quantify the effects of the different types of duplications on the structure of the PPI network.

**Motif enrichment and Z-score.** The standard way to measure network motif enrichment is by reporting the Z-score associated with the motif counts. The Z-score is calculated in the

following way:

$$Z = \frac{n - \bar{n}_{null}}{\sigma_{null}}$$

where $n$ is the motif count in the real data, $\bar{n}_{null}$ and $\sigma_{null}$ are the mean value and the standard deviation of the motif count distribution in the null model. Z-scores are considered to be significant when their absolute value is larger than $\sim 5$. We generated 100 realizations per each of the random models that are defined in a following section.

## Regulatory redundancy and similarity coefficient

As a measure of the interaction similarity between two duplicated genes, we used the Sorensen-Dice Similarity coefficient, defined in the following way for two sets $A$ and $B$:

$$S(A, B) = \frac{2|A \cap B|}{|A| + |B|}. \tag{2}$$

In our case, $A$ and $B$ are the sets of interactions (regulators, targets or PPI depending on the task at hand) of two different genes $a$ and $b$. This measure ranges from 0, when the two genes have no common interactions, to 1, when two genes share all of their interactions. Note that this similarity score is more general than motif enrichment, since we only take into account interactions common to both genes in a couple of paralogues and do not restrict in any way the connectivity between them. In some cases, for example for mixed-type motifs, the definition and interpretation of a similarity score is not straightforward and we resort to the Z-scores to gain more clear insights on the contribution of gene duplication. A more in-depth discussion on the differences between the similarity scores and the motif Z-scores can be found in the corresponding Results section, while a simple graphical example of the similarity between two regulators is shown in Fig 1F.

The statistical significance of the comparison between the similarity distributions of two different categories of gene couples is assessed by means of a two-tail Mann-Whitney U Test, with its associated P-value. The P-values of the comparisons between the real distributions and the null models are reported directly in the figures. If a comparison between two distributions is statistically significant ($P < 0.01$) we show in the figures the following symbols: $^*$ for SSD-WGD comparison, ■ for WGD-NOT DUPLICATED comparison and ▲ for SSD-NOT DUPLICATED comparison. Note that when the symbol is reported, the P-value is typically much lower than the 0.01 threshold, and usually we have at least $P < 1e - 5$.

## Similarity score vs. Z-scores

It is worth noting that the motif enrichment Z-score and the similarity score distributions do not convey the same information. The Z-score counts the overall number of times we encounter a motif in the network, thus generically measuring the contribution of a type of duplicate to the non-random local structure of the whole network and the tendency to retain a specific motif when it is created in the network, either by chance or by other mechanisms (such as gene duplications). It does not, however, convey any information regarding the way in which motifs are distributed among different couples of duplicates, which is instead captured by our similarity measure. This is a very important statistic for our purposes, since we can interpret the similarity score of a duplicate couple as a proxy of the evolutionary constraints that act on it. In fact, higher similarity implies that a stronger evolutionary pressure is preventing the duplicated genes from changing their interactions, and thus their role in the regulatory network. Note that, in principle, the same kind of effect can derive from the duplication age of the

paralogues—younger paralogues did not have enough time to lose or rewire connections and thus share more interactions than older ones. This effect is indeed present and shown in Fig B in S1 Text. We mitigated this kind of bias by considering only SSD couples that were duplicated approximately in the same distant time when also the two rounds of WGD took place, as explained above.

### Null models

We evaluated the motif enrichment by suitably rewiring the regulatory and protein interaction networks. More precisely we constructed randomized versions of the networks using the *degree-preserving* procedure proposed in [43] and illustrated in Fig 1G. This randomization algorithm destroys the local topology of the network but leaves the node degree intact, so that each gene retains the same number of interactions as in the real network, only with different neighbors. In this way we can rule out the possibility that the enrichment patterns we observe are only due to degree-degree correlations in the paralogues, since these correlations are kept unaltered in the ensemble of randomized networks. This is a standard procedure and has also been implemented in widely used motif counting software packages [28, 44].

If the motif under study involves interactions of different types, e.g. transcriptional and protein-protein interactions, we constructed several null models, each one with a randomized version of a different network while keeping the others fixed. Since this work is mainly focused on the effects of duplications at the transcriptional level, we report in the main text only the Z-scores referred to the randomizations of the ENCODE transcriptional regulatory network for mixed-type motifs. The complete results can be found in Fig G in S1 Text.

We also compare the results about interaction similarities of the paralogues with interaction similarities of random non-duplicated gene couples (labelled as "not DUP" in the figures), in order to highlight the role of duplication mechanisms in shaping the network structure.

## Results

The following sections present the results of our motif enrichment analyses in order of increasing topological and functional complexity of the circuits considered.

### Degree distributions

In network theory, the *degree* of a node, which in our case represents a gene, is the number of interactions it has with other nodes in the network. For directed networks, such as transcriptional networks, one can further distinguish between the *in-degree* of a node, i.e., the number of incoming links, and the *out-degree*, i.e., number of outgoing links. The degree distributions of the different networks considered are shown in Fig 2. The degree distributions and the average degree of genes duplicated by SSD and WGD do not display any striking difference with respect to the global degree distributions. Therefore, duplications do not display specific biases in terms of gene degree in the different networks considered. This is a relevant preliminary observation, since in the following we will focus on regulatory circuits whose statistics could be dependent on the degree of the nodes.

### Duplicated genes often interact at the protein level

The first question we address is about the tendency of duplicated genes to interact at the protein level. The PPI network (see the Materials and methods section) is very sparse, with 15,762 nodes and only 237,272 links. In this network, we identified 65,057 SSD pairs and 6,182 WGD

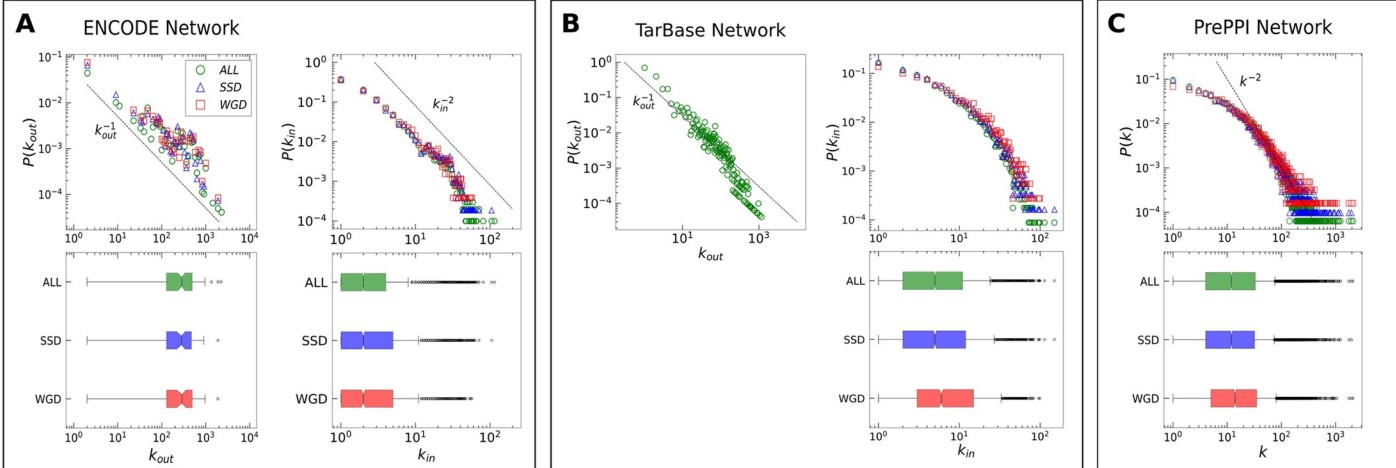

**Fig 2. Degree distributions.** Indegree ($k_{in}$) and outdegree ($k_{out}$) distributions of **(A)** the ENCODE transcriptional regulatory network and of **(B)** the TarBase miRNA-gene regulatory interactions network, and the degree ($k$) distribution of **(C)** the PrePPI protein-protein interactions network. Each degree distribution is shown both as a probability distribution (upper figure) and as a boxplot (lower figure). The global degree distribution of each network is represented in green, while the degree distributions of genes involved in a SSD couple and in a WGD couple are represented in blue and red, respectively. Dotted lines, corresponding to the reported scaling of the degree, are not the result of a fit and are shown as a reference only.

pairs. Among these duplicated genes, approximately 4% of SSD pairs and (17% of WGD pairs show evidence of a protein-protein interaction in the PPI database.

Such percentages, shown in Fig 3, are remarkably high. In the null models used for comparison the proportion of duplicates with an interaction never exceeds 1% and it is usually much lower. This leads to the impressive Z-scores reported in the figure. This behavior is also in stark contrast with the $\sim$ 0.2% of couples of non-duplicated genes with a protein-protein interaction. Overall, we observe a strong correlation between presence of links in the PPI network and the pairing organization of duplicated genes. In other words, duplicated genes have a high probability of interaction in the PPI network. This effect is more pronounced for WGD duplications with almost 1 in 5 couples presenting a protein-protein interaction, compared to just 1 in 25 in the SSD case.

We also analyzed the tendency of couples of duplicated genes to form protein complexes with a third common protein, which is captured by the statistics of co-interaction motifs presented in Fig 4. In particular, Fig 4A shows that WGD couples have a higher interaction similarity with respect to SSD couples and, generally, duplicates have a significantly larger proportion of common interactions than non-duplicated couples. This is confirmed by the comparison with the null model obtained by rewiring the PPI network, as discussed in the Materials and methods section (Fig 4C). This tendency explains the enrichment of co-interaction motifs shown by the Z-scores in Fig 4B.

The evolutionary tendency to retain WGD couples that participate in common protein complexes agrees with previous observations in yeast [26, 45], where the observed tendency was less significant but exactly in the same direction. This result also agrees with a previous observation that proteins belonging to protein complexes were retained more frequently after WGD events than SSD events [46]. The same trend was reported for the human genome using a database of transient protein complexes [22]. We shall see in the Discussion section a nice example (the RAR/RXR pathway) of how the retention of protein-protein interactions among WGD pairs and the tendency to maintain their interactions with common partners may

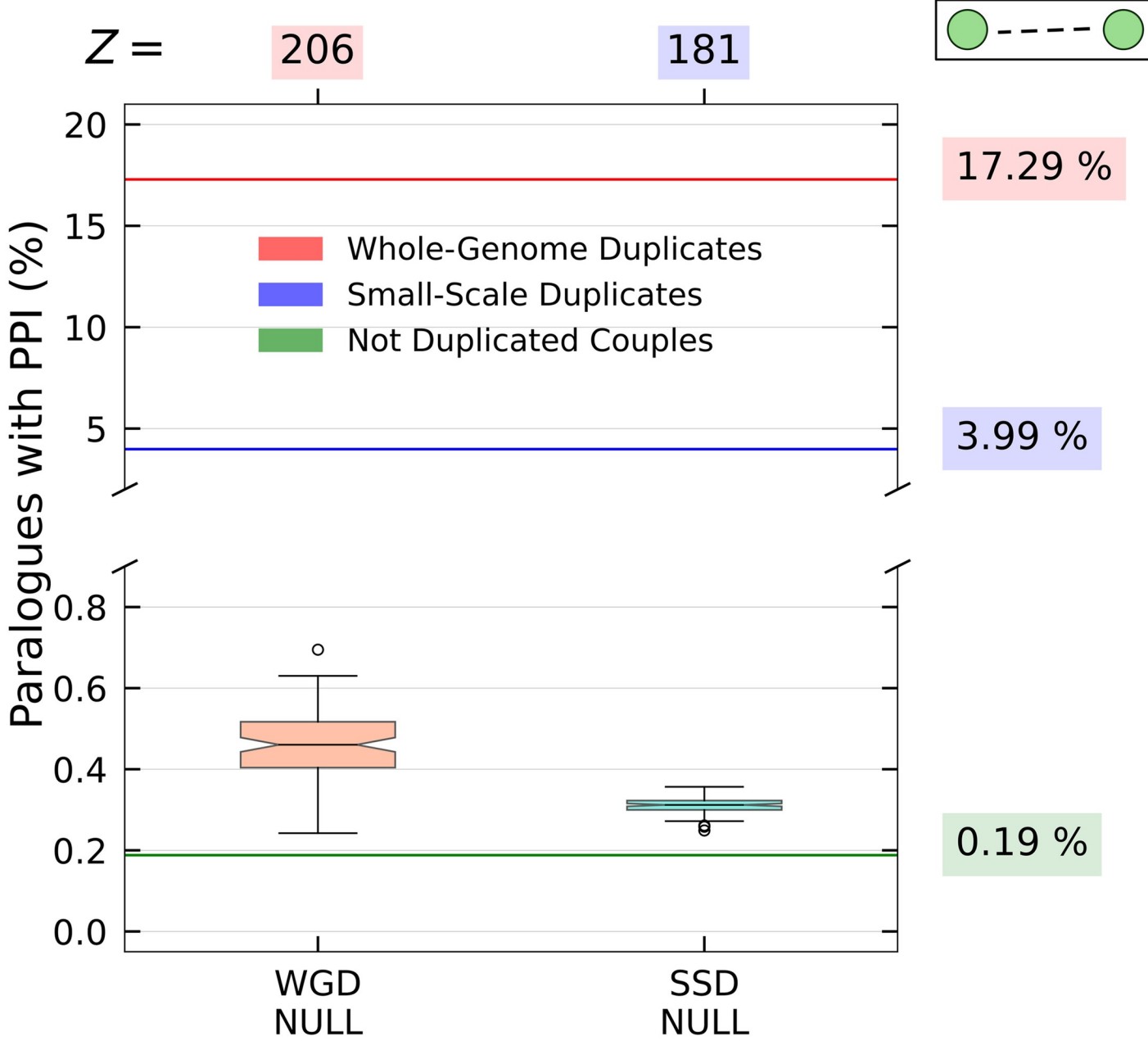

**Fig 3. Interactions of duplicated genes at the protein level.** The percentages of gene pairs that present an interaction in the PrePPI database are indicated by the bold horizontal lines and explicitly stated in the labels on the right. The null model distributions are reported in the boxplots and the corresponding Z-scores are shown at the top.

increase the variety and complexity of the functions performed by the genes involved in the WGD event.

### *V* motifs are enriched of WGD transcription factors

Transcriptional *V* motifs are genetic circuits in which a couple of duplicated transcription factors regulate a common target gene. The motif enrichment analysis and the similarity distributions indicate that WGD pairs of TFs tend to co-regulate the same target genes more than SSD

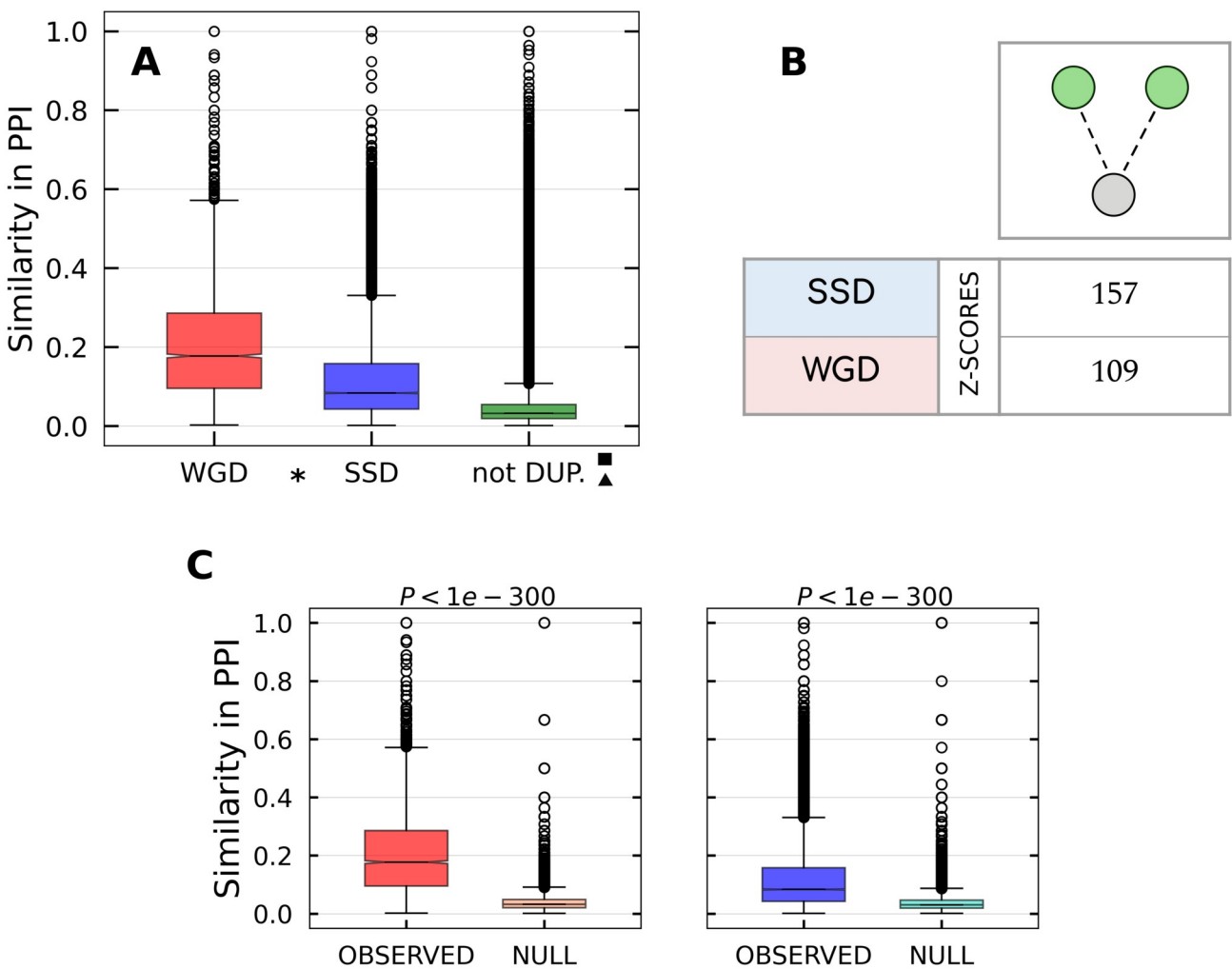

**Fig 4. Pairs of duplicated genes interacting with a third protein.** (**A**) Similarity distributions for WGD, SSD and not duplicated gene couples in the PrePPI network. All of the pairwise comparisons between distributions are statistically significant, as indicated by the presence of the symbols explained in the Materials and methods section. (**B**) Z-scores measuring the enrichment of the co-interaction motif with respect to the null model. (**C**) Pairwise comparison between each real similarity distribution and the null distribution for the respective duplication type.

pairs, whose behaviour is instead comparable with that of non duplicated TF couples (Fig 5A). Since the number of duplicated TFs (both through WGD and SSD events) is rather small, motif enrichment analysis and similarity scores are expected to show larger fluctuations and smaller Z values. However, Fig 5 shows that the result are still consistent. These findings indicate that WGD had a crucial role in shaping the transcriptional regulatory mechanisms, by introducing regulatory redundancies that were retained by evolution over millions of years. On the other hand, regulatory redundancies created by SSD duplications have been generally lost or rewired during evolution. A similar phenomenon was observed in yeast [30], and thus seems to be an universal trend characterizing WGD-derived genes.

The different behavior of WGD and SSD derived couples is corroborated by the observation that WGD pairs of TFs tend to maintain the same DNA Binding Sequence (DBS) much more than SSD pairs. In fact, out of the 25 pairs of WGD TFs, 20 (i.e 80%) kept the same DBS (more precisely they belong to the same motif family, as defined in [47]), while in the SSD case this

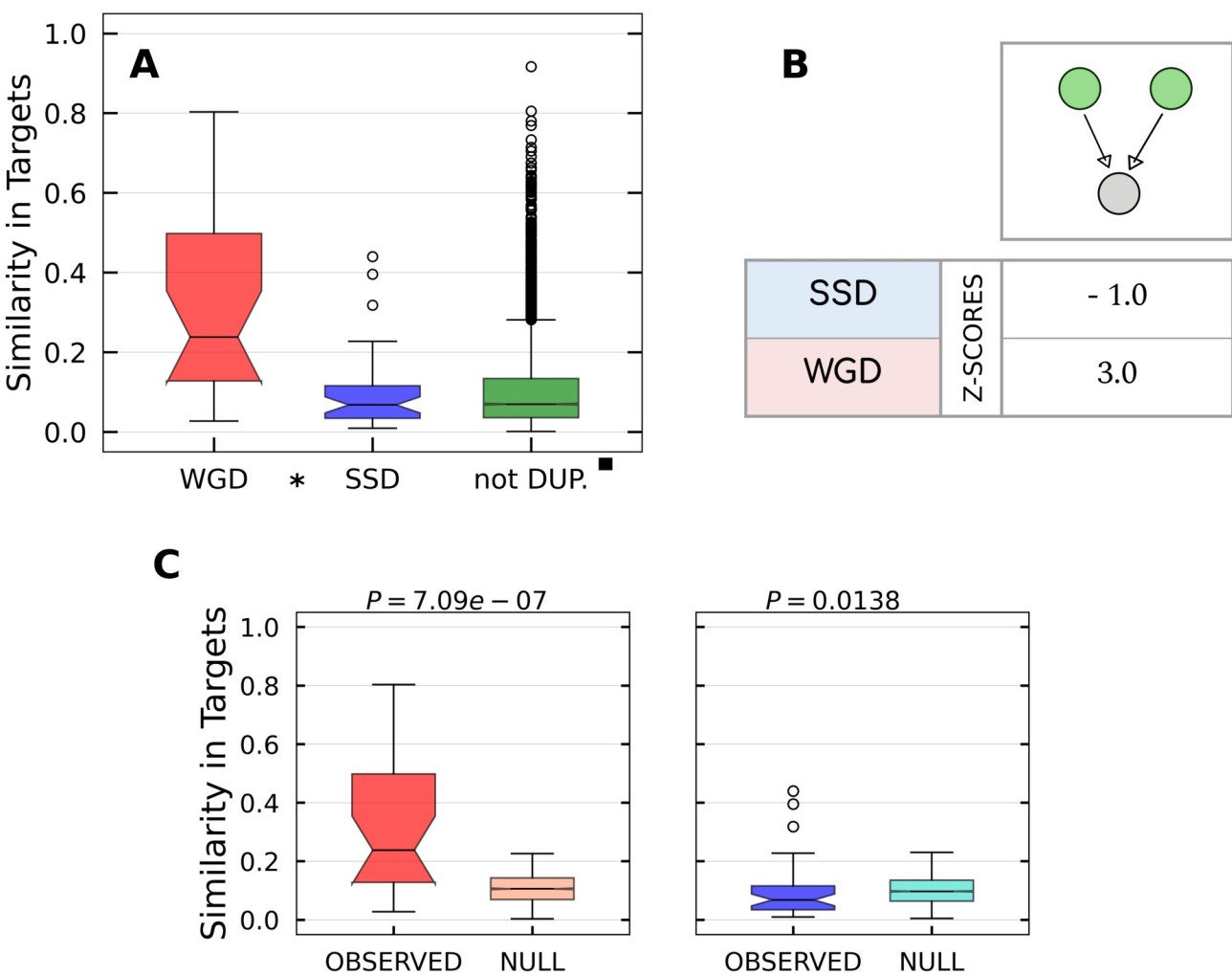

**Fig 5. Transcriptional _V_ motifs. (A)** Similarity distributions for WGD, SSD and not duplicated TF couples in the ENCODE network. As indicated by the presence of the symbols explained in the Materials and methods section, the difference between SSD and not-duplicated distributions is not statistically significant while the comparisons involving the WGD distribution are instead significant. **(B)** Z-scores measuring the enrichment of the _V_ motif with respect to the null model. **(C)** Pairwise comparison between each real similarity distribution and the null distribution for the respective duplication type.

happens only for 7 out of 41 TFs pairs. The specific conservation of DBS in WGD pairs was observed also in yeast [48], thus suggesting that it could be a general phenomenon.

## Λ motifs are enriched in duplicated targets

Λ motifs are simple circuits in which a regulator acts on a couple of targets. We considered transcriptional and miRNA-mediated Λ motifs as reported in Figs 6 and 7 respectively. The similarity distributions of WGD and SSD genes are both larger than the non-duplicate one for both types of Λ motifs. Coherently, the Z-scores indicate enrichment for both SSD and WGD motifs. The Z values suggest that motifs derived from SSD have been retained with higher significance with respect to WGD ones. The same trend is present in miRNA-mediated motifs, but with lower enrichment scores. Overall we observe a tendency of duplicated couples to share the same regulatory interactions. The pattern is more evident at the trascriptional level, and it is stronger for SSD than for WGD pairs.

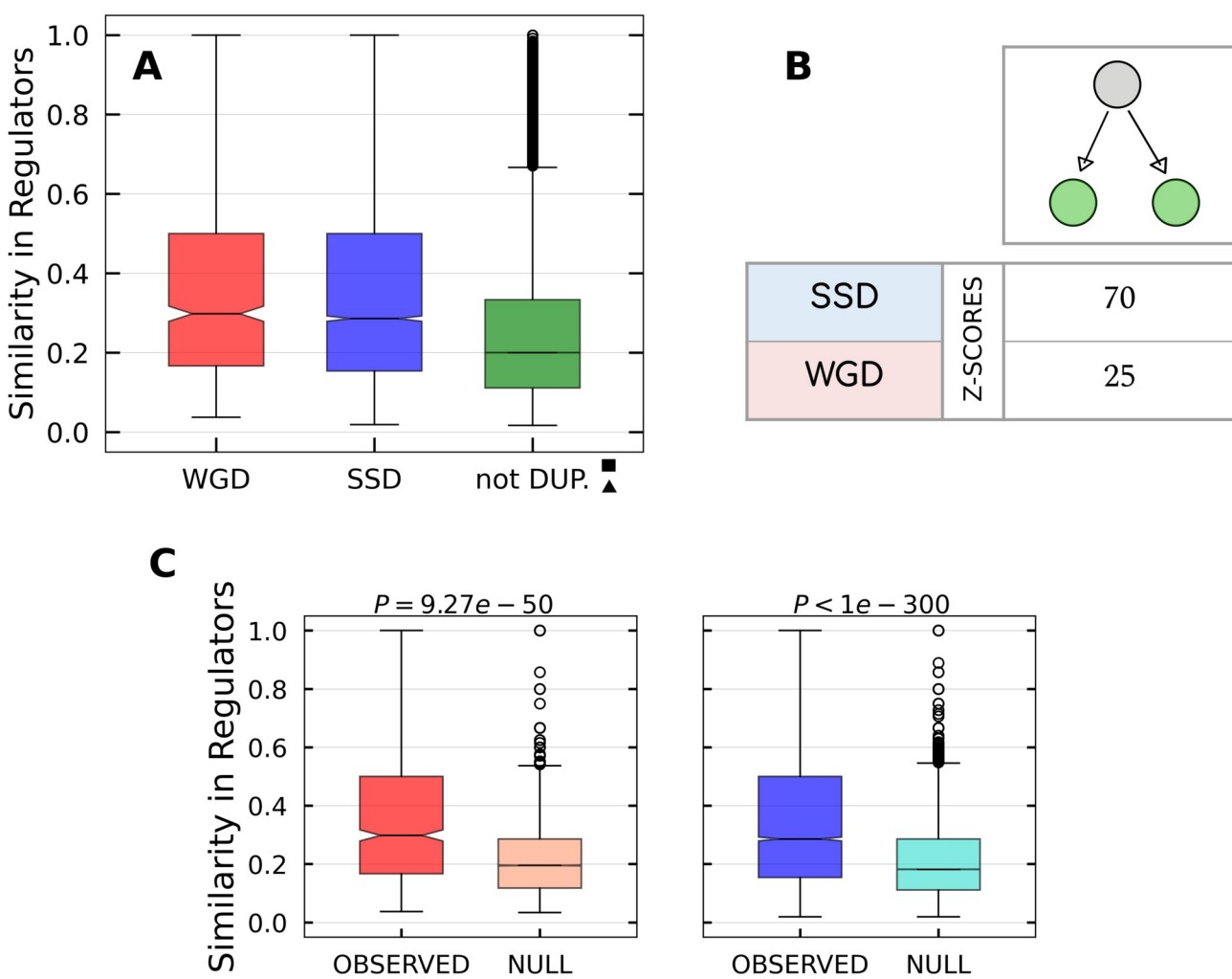

**Fig 6. Transcriptional Λ motifs. (A)** Similarity distributions for WGD, SSD and not duplicated target genes couples in the ENCODE network. As indicated by the presence of the symbols explained in the Materials and methods section, the difference between SSD and WGD distributions is not statistically significant, while both of them are significantly greater that the similarity distribution of non duplicated genes. **(B)** Z-scores measuring the enrichment of the Λ motif with respect to the null model. **(C)** Pairwise comparison between each real similarity distribution and the null distribution for the respective duplication type.

## More complex motifs are enriched in duplicated genes

The role played by WGD-derived genes in shaping the regulatory network emerges more clearly looking at more complex network motifs such as Feed-Back Loops (FBLs), Feed-Forward Loops (FFLs) and BiFan-type motifs (Figs 8 and 9). These motifs were all shown to be associated to relevant specific functions that will be discussed in the corresponding sections.

**FBLs involving pairs of WGD TFs are predominant.** Feedback Loops (FBLs) are a key component of regulatory networks, since they can implement bi-stable switches [29] that represent an excellent decision-making circuit. FBLs can be easily created by duplicating a TF with a self-regulating loop and self regulation is a widespread network motif, from bacteria to humans [29]. This simple motif is associated to several important functions, such as the the modulation of the expression response time, robustness to stochastic noise, and bimodality in the protein levels [29]. In our analysis, the number of observed FBLs is so small that statistical

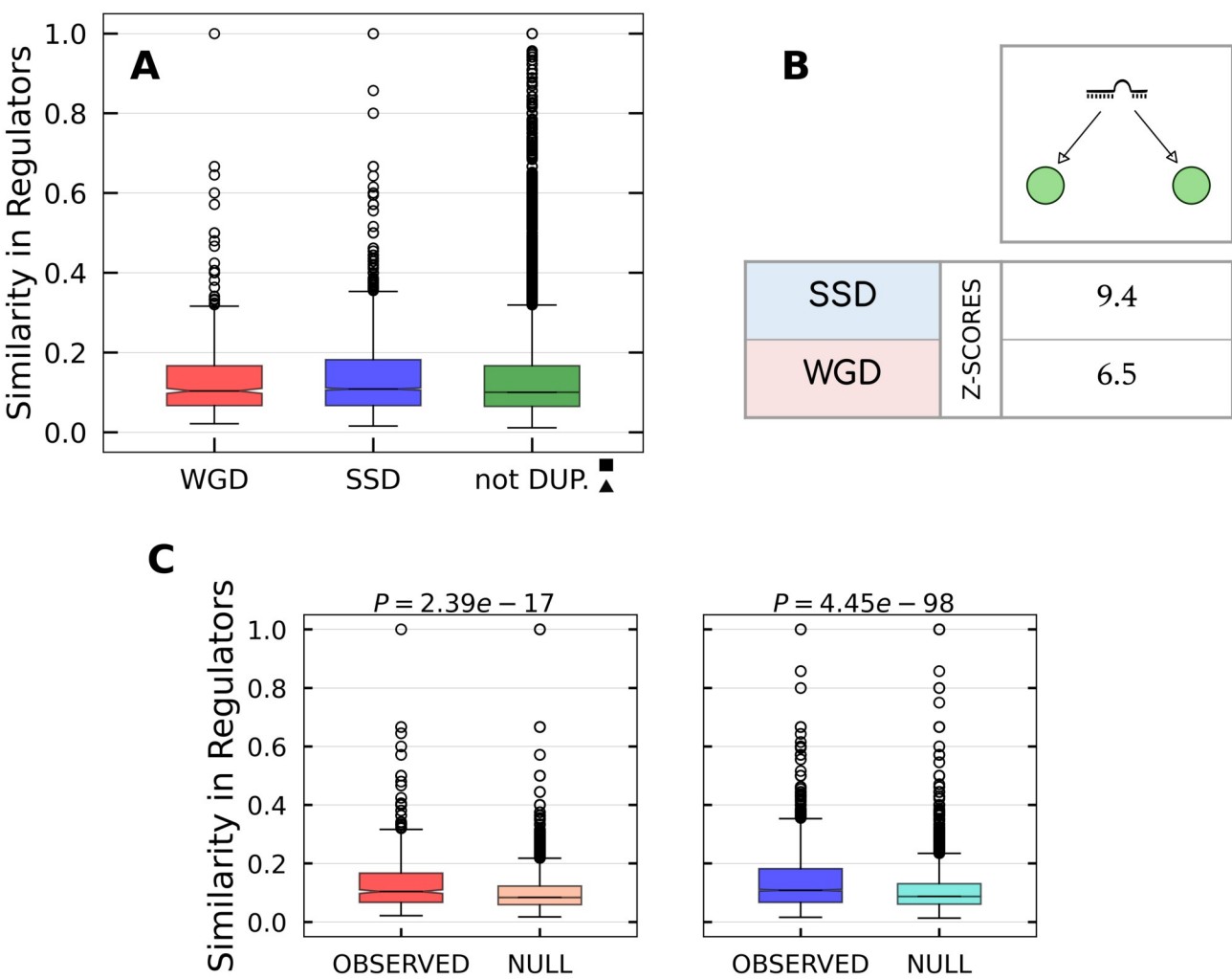

**Fig 7. miRNA Λ motifs.** (**A**) Similarity distributions for WGD, SSD and not duplicated target genes couples in the TarBase network. As indicated by the presence of the symbols explained in the Materials and methods section, the difference between SSD and WGD distributions is not statistically significant, while both of them are significantly greater that the similarity distribution of non duplicated genes. (**B**) Z-scores measuring the enrichment of the Λ motif with respect to the null model. (**C**) Pairwise comparison between each real similarity distribution and the null distribution for the respective duplication type.

tests are not meaningful, thus we simply categorised the 25 pairs of WGD TFs and the 41 pairs of SSD TFs according to their topological configuration. Fig 8 reports the duplicated TF couples that contain at least one gene with a self-loop or that display a mutual regulatory interaction. We immediately see that FBLs involving SSD pairs are completely absent in the network, while 3 out of the 25 pairs of WGD TFs present in the network display a FBL topology and, interestingly, all 3 pairs involved in a FBL motif also present two self-loops. In general, the data presented in Fig 8 show that it is more likely for a pair of WGD-derived TFs to retain a self-regulatory mechanism, together with some kind of mutual regulatory interaction. These observations suggest that the evolutionary pressure favoured the retention of new FBLs created during the two WGD rounds while disfavouring the retention of those created by a SSD event.

**FFLs involving pairs of WGD genes are strongly enriched in the regulatory network.**
Feed-Forward Loops (FFLs) are another fundamental component of gene regulatory networks

| | ONE SELF-LOOP | | TWO SELF-LOOPS | |
|---|---|---|---|---|
| **SSD** **A** | POU2F2 POU5F1<br>GABPA ELK4<br>GABPA ETS1<br>GABPA ELF1<br>GABPA SPI1<br>E2F6 E2F4<br>ESRRA NR3C1<br>FOSL2 ATF3<br>FOSL2 BATF<br>SREBF2 USF2<br>SREBF2 USF1<br>SRF MEF2C<br>BHLHE40 HEY1<br>REST ZNF274 | | | |
| | 14/41 | 0/41 | 0/41 | 0/41 |
| **WGD** **B** | E2F6 E2F1<br>FOSL2 FOSL1<br>ESRRA ESR1<br>JUN JUNB<br>JUND JUNB<br>GATA2 GATA3<br>GATA1 GATA3 | FOSL2 FOS<br>SREBF2 SREBF1<br>MEF2A MEF2C | STAT3 STAT2<br>STAT1 STAT3<br>JUN JUND<br>GATA2 GATA1 | SP1 SP2<br>STAT1 STAT2<br>FOXA1 FOXA2 |
| | 7/25 | 3/25 | 4/25 | 3/25 |

**Fig 8. Feedback loops and self-loops in couples of duplicated transcription factors.** **(A)** SSD and **(B)** WGD duplicate TF couples that contain at least one gene with a self-loop or that display a mutual regulatory interaction in the ENCODE regulatory network, subdivided by equal topological arrangements.

and are often strongly enriched in regulatory networks [29]. Depending on the exact nature and strength of the interactions, they can implement complex functions such as detection of signal persistence, pulse generation, noise buffering and fold-change detection [29].

Fig 9A shows that FFL motifs generated by WGD events are strongly conserved, while the statistics of FFLs involving SSD TFs is compatible with the null model. Once again this clearly

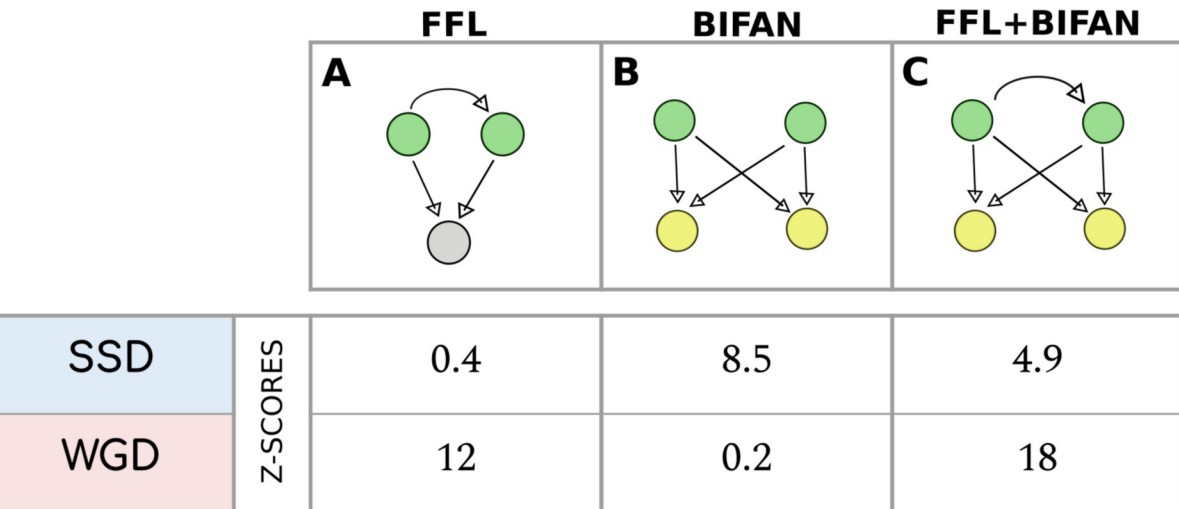

| | | FFL | BIFAN | FFL+BIFAN |
|---|---|---|---|---|
| **SSD** | Z-SCORES | 0.4 | 8.5 | 4.9 |
| **WGD** | | 12 | 0.2 | 18 |

**Fig 9. Transcriptional FFL, Bifan and FFL+Bifan motifs.** **(A)** Transcriptional Feed-Forward Loops (FFLs). **(B)** Transcriptional Bifan motifs (in which no regulatory is present between the two TFs). **(C)** FLL+Bifan motif. In both **(B)** and **(C)** the two regulators and the two targets are duplicated couples of the same type (i.e. both WGD or both SSD pairs).

shows that evolutionary constraints applied to WGD genes are very different from the ones that affect SSD couples.

**Gene duplications shaped Bifan and FFL arrays.**   Bifan and FFL+Bifan motifs (also called "Multi-output Feed-Forward Loops" in the literature) are shown in Fig 9B and 9C respectively. The main function of these motifs is to integrate different input signals, in order to organize the transcription of downstream target genes. They can both be seen as combinatorial decision-making devices, but with an important difference: the additional presence of a regulatory interaction between the two TFs in the second case transforms a simple Bifan into a double FFL, which allows to combine the input signals in a nonlinear fashion, leading to more complex regulatory programs. Another peculiarity of Bifan motifs is their tendency to cluster together, forming extensive superstructures named "Bifan arrays" [48] or "Dense Overlapping Regulons" (DORs) [29], that were identified for the first time in E. Coli [49]. In such superstructures, regulators and targets are arranged on two different layers, with a very large number of regulatory interactions between them. The situation is similar to the one depicted in Fig 1F and 1G, but in real regulatory networks Bifan arrays can involve dozens of genes. The additional presence of regulatory interactions among regulators further increases the complexity of the functions that can be implemented.

We consider the special case where both the regulators and the targets are two—different— duplicated couples, along with motifs that do not contain any duplicated couple. Their levels of enrichment in the ENCODE transcriptional network are shown in Fig 9B for simple Bifans and in Fig 9C for the FFL+Bifan configuration.

The relevance of these two motifs in the structure of the regulatory network is confirmed by their statistical enrichment. In particular, simple Bifans are retained with higher probability when they are created by SSD duplications, while WGD pairs are preferentially involved in FFL+Bifan motifs. This result again confirms that WGD-derived genes are subjected to different evolutionary constraints with respect to SSD-derived genes, and that WGD has driven the formation of motif that are associated to more complex functions.

## Synergy between different layers of regulation is facilitated by duplication events

By analysing different layers of regulation combined together, we can quantify the role of duplication events in fostering the synergy between different regulation layers. For example, considering $\Delta$ motifs we can assess the tendency of a particular type of regulators to act on a couple of duplicated genes that also interact at the protein level (Fig 10A). We observe a strong enrichment of both SSD and WGD motifs, with a slight preference for the former type, which is in line with the results reported in the section on $\Lambda$ motifs. In the case of miRNA-mediated $\Delta$ motifs (Fig 10B), we again observe a clear role of duplicated genes in their retention but there are no clear preferences for SSD or WGD genes.

The enrichment analysis for the mixed-type Bifans in absence of protein-protein interactions, i.e., the motif observed when a duplicated pair is simultaneously involved in a transcriptional and miRNA-mediated $\Lambda$ motif, are reported in Fig 10C. The enrichment of mixed-type Bifans with additional protein-protein interactions between the duplicated genes, is instead shown in Fig 10D. Different types of duplicates appear to promote different integration strategies between layers of regulation. SSD couples are strongly associated with integration between miRNA and transcriptional regulators, when there is no direct PPI interaction between the targets. On the other hand, WGD couples promote the retention also of a direct PPI link between them. This clearly shows that gene duplications facilitate the creation of a significant three-way synergy among the three layers of regulation. This effect can in principle lead to more complex

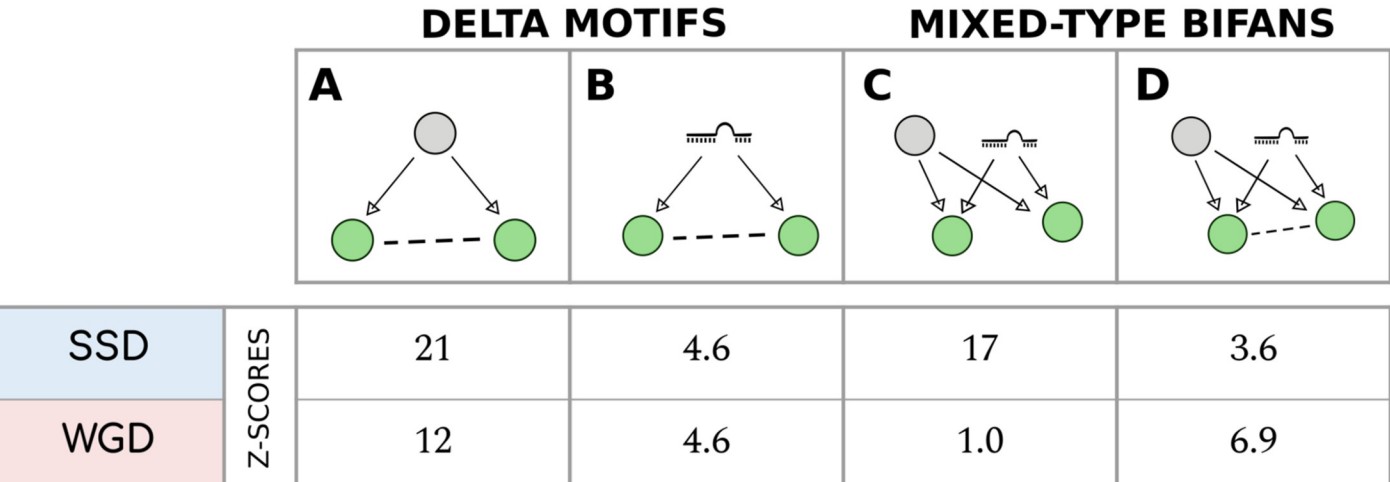

**Fig 10. Motifs with mixed-type regulatory interactions. (A)** Transcriptional Δ motifs. **(B)** miRNA-mediated Δ motifs. **(C)** Mixed Bifan motifs, in which a pair of target genes are regulated both by a common TF and a common miRNA. **(D)** Mixed Bifan motif in which the two target genes interact at the protein level. The reported Z-scores are referred to the null model obtained by randomizing the transcriptional regulatory network (apart from the miRNA Δ motif for which the miRNA-gene network is randomized).

and robust regulatory mechanisms. In fact, the combination of miRNA-mediated and transcriptional regulatory interactions has been shown to ensure optimal noise control, together with a set of interesting complex properties like adaptation and fold-change detection, depending on the parameters of the regulatory interactions [50, 51].

## Discussion

### Target redundancy and dosage balance

The exact mechanisms involved in the retention of duplicated genes are still debated, but most proposed explanations focus on dosage balance constraints [52–54]. For example, a recent analysis of genetic interactions involving WGD couples in yeast proposed that evolutionary trajectories of duplicated genes are dictated by the combination of dosage balance constraints with functional and structural entanglement factors [55]. Another recent study on *A. thaliana* similarly concluded that dosage balance constraints operate immediately after WGD and that duplicate gene retention patterns are shaped by selection to preserve dosage balance [56].

The dosage balance explanation relies on the importance of keeping the correct stoichiometric ratios of protein products within the cell. If the balance is preserved by the duplication event, the duplicated genes will be conserved by evolution with higher probability. This scenario was first proposed to explain the retention of WGD duplicates, since the duplication of the whole genome facilitates an overall balancing of gene expression [54]. This is especially important for classes of genes which show a high level of dosage sensitivity: such genes are preferentially retained in double copy over long evolutionary timescales [57]. Studies conducted on the metabolic network of *A. thaliana* also suggested that different types of dosage constraints—relative and absolute—influence the retention of duplicates at different timescales after WGD events [58].

The dosage-balance principle was also recently invoked to explain SSD retention [59]. In this case, dosage balance (and thus duplicate retention) is granted by a substantial decrease in gene expression of the duplicated pair, which allows to re-balance gene dosage after

duplication. Examples of this last type of behaviour have been found both in yeast and in mammals [59].

The decrease in expression levels needed for dosage balance could be achieved more easily if both duplicated genes were regulated by the same set of TFs, possibly the same TFs which regulated the ancestral gene [59]. The presence of an evolutionary pressure to keep co-regulation of duplicated targets is also supported by recent observations: duplicated gene pairs are enriched for co-localization in the same Topologically Active Domain (TAD), share more enhancer elements than expected, and have increased contact frequencies in Hi-C experiments [60]. From a regulatory network perspective, this evolutionary pressure would imply the selective enrichment we observe of the transcriptional Λ motifs stemming from duplicated targets.

However, this is not the only reason for which one could expect an over-representation of the Λ motif. Motifs of this type ensure a reduction of the relative fluctuations of the two targets [51] and improve the stochastic stability of the duplicated genes. This noise buffering action is particularly effective in presence of a combined and coordinated action of transcription factors and miRNAs [50, 51], i.e., in presence of a "mixed"-type network motifs. All of these considerations are indeed confirmed by the findings presented in Figs 6, 7 and 10C.

Dosage balance constraints and stochastic stability are particularly important if the two duplicated proteins are in interaction between them or are involved in a complex [61]. If this is the case, we should expect a specific enrichment of protein-protein interactions between the two duplicated genes and of Δ motifs. These effects are indeed observed in our analysis (Figs 3, 4 and 10).

The tendency to interact and to share interacting proteins is even more evident for WGD-derived gene couples. This could be again a consequence of how the two different mechanisms of duplication alter the dosage balance [21].

## Regulatory redundancy

It is widely recognized that gene duplications played a central role in the evolution of gene regulatory networks [42, 62] and in setting the TF repertoire [47].

An immediate consequence of TF duplication is the creation of a regulatory redundancy, meaning that after the duplication event the two TFs regulate the same set of target genes. However, this potential functional redundancy is expected to be transient. In fact, during evolution one gene copy may be lost or become a pseudogene, it may acquire a new function (neofunctionalization) [1], or it may share the ancestral functions of the original gene with the other copy (subfunctionalization) [63]. The typical completion time for these processes is of a few millions of years [64], thus for most of the SSD and for all the WGD gene pairs, we should expect no functional redundancy at all. On the contrary, there are strong indications that this is not the case and that for several pairs of both SSD and WGD redundancy is preserved, in some cases, for billions of years [65].

Our study suggests that the retention of regulatory redundancy is strongly dependent on the duplication mechanism. The topological enrichment of *V* motifs an the distribution of target similarity (Fig 5) suggest a significative preference for WGD TF pairs to retain common targets. SSD couples display instead a weak similarity in targets, compatible with null models. Therefore, WGD events seem to have promoted regulatory redundancy during network evolution. Interestingly, there is a non-trivial relation between redundancy in the interactions of the transcription factor repertoire and organismal complexity [47]. This associates once again WGD events to an increased complexity.

There are several possible paths that connect genetic regulatory redundancy with complexity. First of all, regulatory redundancy can increase the robustness against mutations [66],

which is a safety mechanism that is more and more necessary as the interplay of regulatory interactions increase in complexity. Moreover, regulatory redundancy facilitates the implementation of articulated combinatorial regulations. In many cases two duplicated TFs could keep the same set of target genes, but evolve to respond to different cellular signals or to interact with different upstream proteins [2, 67]. We shall see a nice example of this pattern in the RAR/RXR pathway which we discuss below.

In principle, combinatorial regulation—and the associated benefit of an increased environmental responsivity—could evolve by combining the regulations of two TFs, with no need for specifically retaining duplicated TFs. However, such a mechanism would unavoidably increase the noise in the regulatory process. There is indeed a tension between environmental responsivity and noise control in gene regulation, and it has been suggested that it could be resolved by gene duplications [68, 69]. This hypothesis was tested in yeast for the specific Msn2-Msn4 pair of WGD-derived Transcription Factors [68], and our results suggest that it could be a general evolutionary trend that applies also to gene regulation in vertebrates.

Most of the results mentioned above on duplication mechanisms are based on observations and experiments performed in model organisms like S. cerevisiae and A. thaliana. The new data on WGD genes give us the unique opportunity to extend previous studies to encompass the vertebrate lineage. We observed that several trends are conserved across different species and overall it seems that ancient WGD events had a relevant role in shaping current regulatory redundancy.

## FFL and Bifan arrays

The specific combination of FFL+Bifan arrays that, we found, is promoted by WGD-derived genes can have important consequences on the network dynamics. By combining the combinatorics of Bifan with the nonlinear signal integration of FFLs, these circuits can process signals in a highly non-trivial way. As Fig 1E shows, WGD events can create FFL+Bifan motifs in a very easy and natural way. Duplication of a TF with a self-loop interaction generates a couple of TF paralogues with a mutual regulatory interaction and a commmon set of targets. If the original regulator does not have a self-regulatory interaction, the WGD event creates a simple Bifan motif instead. In principle, the same circuits can be generated by a succession of SSD events: the chances of duplicating a TF and its target in two distinct SSD events is reasonably low, but SSD events occur quite frequently. However, there is no guarantee that the created motif will survive. In a relatively short evolutionary timescale many of the created connections could be rewired and duplicated genes could be lost. Therefore, the presence of complex structures retained for more than 500 millions of years is non-trivial and imputable to selective pressure. Interestingly, Fig 9 shows that there are specific retention biases for different circuits depending on the the duplication mechanism at the origin of their formation. Our findings suggest that SSD duplications favoured the formation and retention of the less complex Bifan motif, while WGD duplications are associated to more complex FFL arrays. A similar retention pattern (over-representation of Bifan motifs for duplicated TFs and in particular for WGD versus SSD pairs) was also observed in yeast [48].

These observations again support the conjecture that WGD-derived genes follow a different evolutionary trajectory with respect to SSD ones, and that their emergence favoured the development of complex regulatory strategies.

## Synergy of different layers of regulation

Besides the vertebrates' WGDs, there are other well known examples of WGD events in eukaryotes, such as those observed in *S. cerevisiae* [30, 48] and in *A. thaliana* [10]. Several of

the trends we identified in human are in agreement with previous analysis in those two model organisms, suggesting some universality of the results despite the increase in organism complexity. This increase in complexity is also linked to the presence of several post-transcriptionl layers of regulation, such as miRNA regulation, that are much less developed in simpler organisms such as yeast. Analyzing the human regulatory network, we could identify an important role of gene duplication events in promoting the interplay between different layers of regulation. Specifically, we identified an emergent statistical enrichment of motifs involving both protein-protein interactions and trascriptional regulation, as well as motifs combining transcriptional and post-transcriptional regulation. This agrees with the general observation that complex regulatory functions like adaptation, fine tuning, fold change detection or noise buffering can be better achieved by suitable combinations of miRNAs and TFs, arranged in well defined network motifs [50, 51, 70]. Our analysis indicates that several of these mixed motifs arose with ancient gene duplication events—both SSD and WGD—at the beginning of the vertebrate lineage and were then conserved by evolution for more than 500 million years.

## An example of WGD importance: The RAR/RXR pathway

In this section we discuss more in depth the RAR/RXR pathway (schematized in Fig 11A and 11B), a tangible example that hints at the importance of WGD events in contributing to the evolution of complex traits in vertebrates. The pathway is composed by four sets of WGD-derived genes: the RAR family (RARA,RARB,RARG), the RXR family (RXRA,RXRB,RXRG), the NCOA family (NCOA1,NCOA2,NCOA3) and the NCOR family (NCOR1,NCOR2). They densely interact among themselves at the protein level (see Fig 11C) and they co-interact with a host of other genes. The choice of this particular example is due both to its central role in the embryonic development of vertebrates and to the high statistical significance of the number of common interactions of the genes involved.

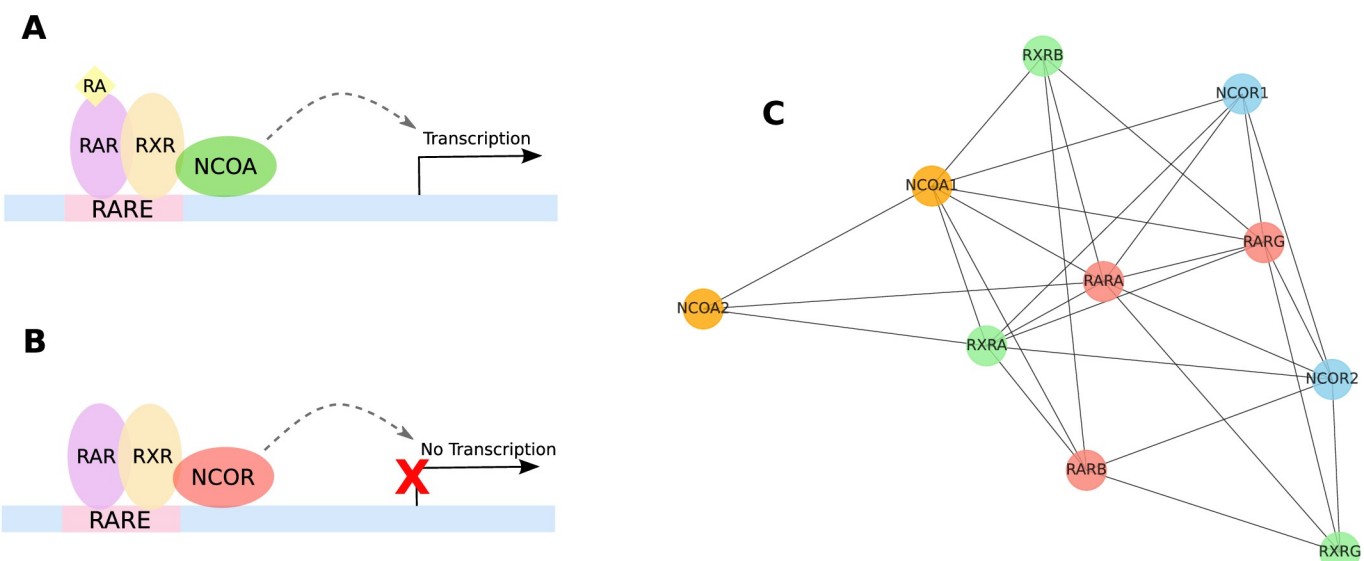

**Fig 11. RAR/RXR pathway is an example of WGD importance. (A)** In presence of Retinoid Acid (RA) the RAR/RXR complex recruits Nuclear CoActivators (NCOA), activating the transcription of the downstream gene. **(B)** In absence of RA the RAR/RXR complex recruits Nuclear CoRepressors (NCOR) instead, blocking the transcription of the downstream gene. **(C)** Protein-protein interactions among the genes involved in the RAR/RXR pathway, common interactions with other genes are not shown for clarity. Genes colored in the same way are WGD copies of a common ancestral gene.

The Retinoic Acid Receptors (RAR) genes are nuclear receptors of the Retinoic Acid (RA) which is a metabolite of retinol (vitamin A). They form heterodimeric complexes with the Retinoid X Receptors (RXR), which then target a DNA binding sequence known as Retinoic Acid Response Element (RARE) and act as transcriptional regulators for a host of target genes. Binding of the RAR/RXR complex at the RARE site induces the recuitment of either the Nuclear Receptor CoActivators (NCOA), in presence of the retinoic acid (Fig 11A), or the Nuclear Receptor CoRepressors (NCOR), when RA is absent (Fig 11B), thus directly activating or repressing the transcription of the target genes. The RAR pathway is known to be involved in the formation of the body axis and is essential for the development of several organs including the hindbrain, the spinal cord, the skeleton, the heart, the eye, the pancreas, the lung and the reproductive tract. Importantly, it plays a prominent role in the development of the central nervous system. It mediates the anteroposterior regionalization, by regulating the transcription of Hox genes and subsequently stimulating neurogenesis and promoting neuronal differentiation. For an in-depth review of the RAR/RXR pathway see for instance [71] and references therein.

The three components of the RAR family, *RARα* (RARA), *RARβ* (RARB) and *RARγ* (RARG), happen to be WGD copies of an ancestral RAR gene. Such ancestral gene can still be found in several non-chordate organisms like, for instance, anellids and mollusks [72, 73]. It has been shown that the ancestral RAR has a much lower affinity with its ligand with respect to the vertebrate RARs [72, 73] and that each of the three WGD copies of RARs evolved to gain a different ligand specificity and expression pattern [74]. At the same time, we observe a large number of common interactions among the three, sign that a significant regulatory redundancy has nonetheless been retained.

The scenario which emerges from these observations (for a thorough discussion see for instance [73]) is that before the WGD event the ancestral RAR was only involved in neuronal differentiation, with no involvement in spatial patterning. After WGD on the other hand, thanks to the higher affinity with the ligand and to the specificity of the binding interactions, the RAR system developed the ability of reading the spatial distribution of the RA. In particular the RAR pathway became, via the regulation of the Hox genes, the controller of the anteroposterior patterning in chordates. Evidently, this important gain of functionality is connected to the two rounds of WGD that created redundant copies of the genes involved in the pathway.

There is one last interesting fact to notice in connection to our discussion on the role of WGD events in the evolution of the RAR/RXR pathway. The anteroposterior patterning must be ultimately due to an increased complexity of the spatial distribution of the RA, otherwise the increased ability of the RAR system to read the RA distribution would have been useless. Such non trivial spatial organization requires an articulated degradation machinery for the RA. This degradation is performed in vertebrates by the CYP26 family, which is also composed by a triplet of WGD-derived paralogues, namely the CYP26A1, CYP26B1, CYP26C1 genes [75]. This fact, once again, strongly points to a fundamental role of WGD duplications in the evolution of some complex vertebrate traits.

## Robustness of the results

The nature of the motifs that we studied and the type of enrichment in which we are interested (WGD versus SSD, or pairs of duplicated genes versus non-duplicated ones) requires a careful control over possible spurious effects. The first necessary control is that the three gene classes do not differ significantly in the number of interactions they have since this could affect the motif statistics. The absence of this possible bias is tested in Fig 2.

To further assess the robustness of our analysis, we considered two alternative protein-protein interaction networks (the PrePPI and STRING-DB network) and two alternative miRNA-gene networks (the TarBase and the mirDIP network). Despite significant differences both in the genes and in the interactions in the different databases, we found consistent enrichment patterns (see S1 Text).

## Conclusions

Gene duplications played a crucial role in the evolution of the human genome, and it is by now widely accepted that two rounds of whole genome duplication happened at the origin of the vertebrate lineage [1]. How these two global-scale events affected the gene regulatory networks is, however, still to be fully understood. Thanks to the recently published lists of WGD pairs [18, 20, 21], we had the possibility to tackle this problem. This paper quantifies the effects of WGD and SSD events on the structure of regulatory networks in human, and the results support the idea that these networks were significantly shaped by the two rounds of WGD at the beginning of the vertebrate lineage.

Our analysis of network motifs specifically indicates that the two rounds of WGD contributed substantially to the overall regulatory redundancy, promoted synergy between different regulatory layers, and typically generated motifs that can be associated with complex functions.

## Supporting information

**S1 Text. Robustness of the results.** In-depth analysis of the possible biases introduced by different paralogue duplication ages and additional validation analyses carried out on alternative PPI and miRNA-gene interaction networks. **Fig A**. Age distribution of WGD and SSD paralogues contained in the interaction networks. **Fig B**. Effects of different duplication ages on the similarity distributions in the different regulatory networks considered in this work. **Fig C**. Degree distributions of the mirDIP (miRNA-gene) and STRING-DB (protein-protein interaction) networks. **Fig D**. Interactions between duplicated genes at the protein level in the STRING database. **Fig E**. Pairs of duplicated genes interacting with a third protein in the STRING database. **Fig F**. Pairs of duplicated genes regulated by the same miRNA the mirDIP database.**Fig G**. Enrichments of motifs with mixed-type regulatory interactions, for different network combinations.
(PDF)

## Author Contributions

**Formal analysis:** Francesco Mottes, Chiara Villa.

**Investigation:** Francesco Mottes, Chiara Villa.

**Software:** Francesco Mottes.

**Supervision:** Matteo Osella, Michele Caselle.

**Visualization:** Francesco Mottes.

**Writing – original draft:** Francesco Mottes, Michele Caselle.

**Writing – review & editing:** Francesco Mottes, Chiara Villa, Matteo Osella, Michele Caselle.

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
