## [Decision Letter · Decision Letter 0]

4 Oct 2021

Dear Mr Mottes,

Thank you for submitting your manuscript "The impact of whole genome duplications on the human gene regulatory networks" for consideration at PLOS Computational Biology. As with all papers reviewed by the journal, your manuscript was reviewed by members of the editorial board and by several independent reviewers. The reviewers appreciated the attention to an important topic. Based on the reviews, we are likely to accept this manuscript for publication, providing that you modify the manuscript according to the review recommendations.

Sincerely,

Yamir Moreno

Associate Editor

PLOS Computational Biology

Jason Papin

Editor-in-Chief

PLOS Computational Biology

[LINK]

Reviewer's Responses to Questions

**Comments to the Authors:**

Reviewer #1: The impact of whole genome duplications on the human gene regulatory networks.

Using human paralogues, this work compares those originating from 2 ancestral Whole Genome Duplications (WGD), which predate the origin of vertebrates, with those originating from Small Scale duplications of similar age. The analysis uses 3 kinds of networks (transcription regulatory networks, protein-protein interaction networks, and miRNA interaction networks). Common units of interaction or regulation are extracted as "network motifs". The main result is that some classes of motifs are differentially enriched, indicating that the two types of duplications play different roles in the evolution of regulation. The authors further discuss how the different motifs can arise, lead to different types of regulation, and can interact to create more complex regulatory structures.

The analysis is similar to a previous one that studied another WGD in yeast. However, this is the first one to study WGD in vertebrates and has significant consequences to our understanding of regulatory networks in humans.

In this manuscript, the problem is clearly stated, due credit is given to previous work, and the discussion of the subject is enriched with new insights.

The datasets seem well thought and clean enough. The authors were careful enough to only include duplications of comparable age. One of their first results was that the connectedness (the node degree) was similar for all the datasets compared. This is important because the statistic could be affected by the degree of the nodes. Controls, statistical methods, and null models are correct. In most results obtained, the Z-scores indicate extremely high significance.

In my opinion, the work is solid and sufficient. I would have liked to see some discussion of specific genes given as examples. That would make the implications of the study more tangible to many readers.

The authors have set a GitHub repository to provide access to their data. One dataset that is missing there, and also not adequately described in Materials and Methods is that of the non-duplicated genes.

The manuscript is well-organized and is easy to follow. Figures are clear easy to understand.

Reviewer #2: Review of Mottes et al.

The paper of Mottes et al. describes the long-term effects of two rounds of Whole Genome Duplication (WGD) at the dawn of vertebrate evolution on the architecture of the human gene regulatory networks. The authors integrated information on transcriptional regulation, miRNA regulation, and protein-protein interactions to analyse the role of both small and large-scale (whole genome) duplications on the structural properties of biological and regulatory networks. The authors conclude that both ancient WGD events played a substantial role in increasing the overall complexity of the vertebrate regulatory network by enhancing its combinatorial organization, with potential consequences on overall robustness and increase in biological complexity.

I have read this paper with great interest. I think the paper is well written and, in my opinion, does a great job in introducing and explaining the aims of the study.

I have gone over the paper twice and could not come up with any major objections. As far as I can judge, the methodology and approach used warrants the conclusions and the authors have been careful not to overinterpret the results. All data are available and have been well described.

Nevertheless, I do have the following comments.

On lines 521-522, the authors state that: “Most of the results mentioned above on duplication mechanisms are based on observations and experiments performed in simple model organisms like S. cerevisiae and A. thaliana. The new data on vertebrate WGD genes give us the unique opportunity to extend previous studies in a more complex setting.” I believe the authors are wrong in assuming that A. thaliana is a simple(r) model organism. A. thaliana has more genes than human (I admit this does not say or mean much), but more importantly, A. thaliana has undergone and survived several whole genome duplications, probably more than any other organism, including human. So, it is not clear to this reviewer why the authors refer to A. thaliana as a simple(r) model organism? Also, in this respect, I have the feeling the authors should perhaps pay more attention to the (recent) literature on whole genome duplications in plants. Apart from these two ancient WGDs in vertebrates, and an additional one, about 300 mya, in the fish lineage, the great majority of genome duplications have been described for plants. For instance, although the authors do mention Arabidopsis a few times (but without going into detail, and often together with the, indeed, simpler yeast), in general I have the feeling that the authors neglect a little too much what is known about plants and WGDs. For instance, the authors write, on lines 455-457: “In this case, dosage balance (and thus duplicate retention) is granted by a substantial decrease in gene expression of the duplicated pair, which allows to re-balance gene dosage after duplication. Examples of this behaviour have been found both in yeast and in mammals.”. There are several recent papers discussing dosage, gene balance, epigenetic remodeling, subgenome dominance etc. in plants as well, and probably (much) more than what has been described for animals and even yeast.

The authors conclude that “… and it is by now widely accepted that two rounds of whole genome duplication happened at the origin of the vertebrate lineage [1]. How these two global-scale events affected the gene regulatory networks is, however, still to be fully understood. Thanks to the recently published lists of WGD pairs [14, 16, 17], we had the possibility to tackle this problem. This paper quantifies the effects of WGD and SSD events on the structure of regulatory networks in human, and the results support the idea that these networks were significantly shaped by the two rounds of WGD at the beginning of the vertebrate lineage. Our analysis of network motifs specifically indicates that the two rounds of WGD contributed substantially to the overall regulatory redundancy, promoted synergy between different regulatory layers, and typically generated motifs that can be associated with complex functions.” While I agree with this overall statement of the authors, I was hoping to see some more speculation on how this ‘overall regulatory redundancy, synergy between different regulatory layers, and generated motifs associated with complex functions’ might have ‘helped’ or facilitated the evolution of vertebrates in particular. Can these WGDs therefore be indeed linked to an increased complexity of vertebrates which likely would not have been possible without these WGDs, as has been suggested by Ohno (1970), or alternatively, could these WGDs and their effects on gene regulatory networks have reduced the risk of extinction as suggested by Crow and Wagner (2006), for instance. Although I understand this is not self-evident, a little more speculation on the possible biological or evolutionary consequences of the specific observations made in this study would be nice, in my opinion. The authors mention some genes (see for instance Fig. 8), part of some pathways, but a deeper discussion on for example how some specific gene(s) and their recruitment in a specific duplicated motif or pathway could have been important for vertebrate adaptation of evolution would be very interesting.

The authors discuss motif enrichment in simple and duplicated networks, but I was wondering whether, for instance in duplicated networks, they have also considered underrepresentation of certain motifs? One could imagine that certain motifs, when duplicated, could be detrimental or would lead to maladaptation or lower fitness. Is this something the authors looked at or considered?

**Have the authors made all data and (if applicable) computational code underlying the findings in their manuscript fully available?**

Reviewer #1: Yes

Reviewer #2: Yes

PLOS authors have the option to publish the peer review history of their article (what does this mean?). If published, this will include your full peer review and any attached files.

Reviewer #1: No

Reviewer #2: No

Figure Files:

Data Requirements:

Reproducibility:

References:

---

## [Editor Report · Decision Letter 1]

12 Nov 2021

Dear Mr Mottes,

We are pleased to inform you that your manuscript 'The impact of whole genome duplications on the human gene regulatory networks' has been provisionally accepted for publication in PLOS Computational Biology.

Best regards,

Yamir Moreno

Associate Editor

PLOS Computational Biology

Jason Papin

Editor-in-Chief

PLOS Computational Biology

---

## [Editor Report · Acceptance letter]

1 Dec 2021

PCOMPBIOL-D-21-01527R1 

The impact of whole genome duplications on the human gene regulatory networks

Dear Dr Mottes,

I am pleased to inform you that your manuscript has been formally accepted for publication in PLOS Computational Biology. Your manuscript is now with our production department and you will be notified of the publication date in due course.

With kind regards,

Zsofia Freund
